# Child defecation and feces management practices in rural Bangladesh: Associations with fecal contamination, observed hand cleanliness and child diarrhea

Mahfuza Islam[1]*, Mahbubur Rahman[1], Leanne Unicomb[1], Mohammad Abdullah Heel Kafi[1], Mostafizur Rahman[1], Mahfuja Alam[1], Debashis Sen[1], Sharmin Islam[1], Amy J. Pickering[2], Alan E. Hubbard[3], Stephen P. Luby[4], Benjamin F. Arnold[5], John M. Colford, Jr.[3], Ayse Ercumen[6]

1 Environmental Intervention Unit, Infectious Disease Division, icddr,b, Dhaka, Bangladesh, 2 Civil and Environmental Engineering, Tufts University, Medford, MA, United States of America, 3 Division of Epidemiology and Biostatistics, University of California, Berkeley, CA, United States of America, 4 Woods Institute for the Environment, Stanford University, Stanford, CA, United States of America, 5 Francis I. Proctor Foundation, University of California, San Francisco, CA, United States of America, 6 Department of Forestry and Environmental Resources, North Carolina State University, Raleigh, NC, United States of America

* mi_sheuli@icddrb.org

**Data Availability Statement:** All relevant data are within the manuscript.

## Abstract

Child open defecation is common in low-income countries and can lead to fecal exposure in the domestic environment. We assessed associations between child feces management practices vs. measures of contamination and child diarrhea among households with children <5 years in rural Bangladesh. We visited 360 households quarterly and recorded caregiver-reported diarrhea prevalence, and defecation and feces disposal practices for children <5 years. We examined caregiver and child hands for visible dirt and enumerated *E. coli* in child and caregiver hand rinse and stored drinking water samples. Safe child defecation (in latrine/potty) and safe feces disposal (in latrine) was reported by 21% and 23% of households, respectively. Controlling for potential confounders, households reporting unsafe child defecation had higher *E. coli* prevalence on child hands (prevalence ratio [PR] = 1.12, 1.04–1.20) and in stored water (PR = 1.12,1.03–1.21). Similarly, households reporting unsafe feces disposal had higher *E. coli* prevalence on child hands (PR = 1.11, 1.02–1.21) and in stored water (PR = 1.10, 1.03–1.18). Effects on *E. coli* levels were similar. Children in households with unsafe defecation and feces disposal had higher diarrhea prevalence but the associations were not statistically significant. Our findings suggest that unsafe child feces management may present a source of fecal exposure for young children.

## Introduction

The proportion of the world population reporting they practiced open defecation fell from 24% in 1990 to 13% in 2015 [1]. In Bangladesh, only 2% of households had no access to a toilet

**Funding:** 'This study was funded by National Institutes of Health (NIH) grant number R01HD078912.The funder had no role in study design, data collection and analysis, decision to publish, or preparation of the manuscript'.

**Competing interests:** The authors have declared that no competing interests exist.

and 4% lacked latrine access in rural areas as of 2013 [2]. However, several recent sanitation trials have shown mixed impact from latrine provision on health outcomes [3–7] and studies that measured fecal contamination at potential household exposure points found little or no effect of sanitation interventions in reducing fecal indicator bacteria [8–10], suggesting other sources of fecal contamination that are not adequately eliminated by typical sanitation interventions. One potential source is child open defecation, which remains common in low-income countries. Despite widespread latrine access, Bangladesh has the second lowest levels of reported safe disposal of child feces in the South Central Asia region [11]. Poor child feces management could be a potential contributor to health risk as young children with poorly developed immune systems have higher incidence of enteric infections than other age groups [12] and their feces are also more likely to contain higher quantities of transmissible pathogens [13]. The presence of a latrine may therefore not necessarily minimize exposure to fecal-oral pathogens through child feces [14], especially for young children who primarily spend time in the home environment and have frequent hand contact with feces or with soil contaminated by feces [15].

Fecal-oral pathogens are transmitted through a variety of routes from one host to the next, either as a result of direct transmission through contaminated hands, or indirect transmission via contamination of drinking water, food, and fomites [16, 17]. Young children frequently place their hands in their mouths, and in Bangladesh, it is also common to eat and to be fed by hand [18]. Previous studies in Bangladesh demonstrated that caregiver's and children's hands can contain fecal indicator organisms at concentrations of >100 colony forming units (CFU) per two hands [19]. The presence of child feces in the household environment could be a potential contributor to fecal contamination of hands in this setting. Drinking water in rural Bangladeshi households also often contains fecal indicator bacteria. While contamination levels are often low at the source (primarily tubewells), the microbiological water quality deteriorates significantly during storage and handling at home [20, 21]. The presence of child feces in the domestic environment could contribute to fecal contamination of tubewell water through infiltration and of stored drinking water via contaminated containers, hands and fomites during collection, handling and storage.

Few studies to date [22–26] have assessed how child defecation and child feces management practices affect contamination along fecal-oral transmission pathways such as drinking water and hands. Understanding the impact of child defecation and child feces management practices on fecal exposure pathways could be important to identify sources of transmission that are not interrupted by conventional sanitation programs and might benefit from targeted interventions. In this study, among households with children <5 years in rural Bangladesh, we aimed to assess the association between reported child defecation/child feces disposal practices and (1) *E. coli* contamination of child and caregiver hands and stored drinking water, (2) observed cleanliness of caregiver and child hands, and (3) child diarrhea.

## Materials and methods

### Study design

We conducted a longitudinal study within a randomized controlled trial in rural Bangladesh (WASH Benefits Bangladesh trial, ClinicalTrials.gov NCT01590095). The parent trial was conducted in the Gazipur, Kishoreganj, Mymensingh and Tangail districts of rural Bangladesh [27, 28]. The trial randomly assigned geographically pair-matched clusters of pregnant women to water, sanitation, hygiene and nutrition intervention vs. control arms and followed their birth cohort of "index children" (children of enrolled pregnant women that were in utero at the time of enrollment) for approximately two years to assess intervention impacts on child

growth, diarrhea and enteric infections. Additional details of the study design and interventions have been described elsewhere [27–29].

We conducted a longitudinal sub-study of environmental contamination among randomly selected households enrolled in the sanitation and control arms of the WASH Benefits trial to leverage the design and infrastructure of the large-scale randomized controlled trial. Households were eligible for enrollment in the sub-study if the index child was alive and available or if there was another child <24 months available in the household. In this analysis, we report measurements from the 360 households enrolled in the control arm of the longitudinal sub-study to assess the relationship between child feces management practices and fecal contamination in the domestic environment.

## Data collection

We visited households enrolled in the sub-study approximately every three months for a total of eight visits over 2.5 years between June 2014 and December 2016. At each visit, trained field staff from the International Centre for Diarrhoeal Disease Research, Bangladesh (icddr,b) used a structured questionnaire to record caregiver-reported defecation and feces disposal practices and 2-day and 7-day prevalence of diarrhea (defined as ≥3 loose stools in 24 hours) for children <5 years. The questionnaire also included information on reported water treatment practices, and the field staff conducted spot check observations to observe drinking water storage containers, household hygiene conditions, sanitation facilities, and the presence of any human feces within the compound. At each visit, field workers also examined caregiver and index child hands (finger nails, finger pads and palms of each hand) and recorded the presence of dirt using a three-point scale (visible dirt particles, unclean appearance, clean). Visible dirt particles were defined as specks of dirt, mud, soil, ash or any other visible material; unclean appearance was defined as no visible dirt particles but general uncleanliness; and clean was defined as would appear after someone washes hands or takes a bath.

## Sample collection

At each visit, field staff collected 250 mL of drinking water from household storage containers by asking participants to provide a glass of water that they would give their young children to drink and pour it into a sterile Whirlpak bag (Nasco Modesto, Salida, CA). If the caregiver provided water directly from the water source, we collected a sample from the household's primary drinking water storage container. Index child and mother hand rinse samples were collected by massaging and shaking the hands, one at a time, in 250 mL of sterile water in a sterile Whirlpak bag. All samples were placed on ice and transported to the icddr,b field laboratory for analysis for *E. coli* within 12 hours of collection.

## Sample processing

Samples were processed with the IDEXX Quantitray-2000 system. Stored drinking water was analyzed undiluted in 100 mL aliquots; caregiver and child hand rinse samples were diluted 1:2 by adding 50 mL of distilled water to 50 mL from hand rinse samples for a total volume of 100 mL. Colilert-18 media was added to samples, followed by incubation at 44.5 °C for 18 hours to enumerate the most probable number (MPN) of *E. coli* [30]. MPN values were derived from the number of fluorescent wells on the trays using the IDEXX Quantitray-2000 MPN table and reported per 100 mL for water samples and per 2 hands for caregiver and child hands. Trays exceeding the upper detection limit of 2419 MPN were classified as too numerous to count (TNTC); the Quantitray-2000 system with this high detection limit was chosen to capture a range of contamination levels.

## Quality control

Ten percent field blanks (one blank for every 10 samples) and 5% replicates (repeat aliquots from the same sample) were processed for quality control. Field workers collected field blanks by asking respondents to pour distilled water from a sterile bottle into a Whirlpak as if collecting a stored water sample and by opening and messaging a Whirlpak prefilled with distilled water as if sampling a hand. One laboratory blank was processed per lab technician per day.

## Data analysis

For our exposure variables, we defined safe child defecation as reported defecation in a potty or latrine, and unsafe defecation as defecation on a piece of cloth, on the floor/bed inside the house, on the ground in the compound/front yard, or in bushes/fields for the child's last reported defecation event; defecation in a cloth was considered unsafe defecation as this does not sufficiently isolate child feces from the environment. We defined safe child feces disposal as caregivers reporting that feces were put/rinsed into latrine or specific pit or buried, and unsafe child feces disposal as feces put/rinsed into a drain, ditch, bush or garbage heap or left on the ground for the child's last reported defecation event. For our outcome variables, we defined *E. coli* prevalence as the detection of $\geq 1$ MPN *E. coli* per 100 mL of drinking water and per 2 hands, and we also calculated $\log_{10}$-transformed *E. coli* counts. We replaced *E. coli* counts over the detection limit with 2420 MPN and counts for non-detects with 0.5 MPN before calculating the logarithm. We defined dirty hands as those with visible dirt particles on palms, pads or nails of one or both hands.

We compared the prevalence and $\log_{10}$-transformed concentration of *E. coli* in stored water and hand rinse samples, the prevalence of caregiver and child hands with visible dirt, and the prevalence of diarrhea with 2-day and 7-day symptom recall periods between households with unsafe vs. safe child defecation and child feces disposal practices. We estimated the prevalence ratio (PR) for the binary outcomes and the difference in log-transformed *E. coli* counts, using pooled data from all follow-up visits. We conducted bivariate and multivariable analyses using generalized linear models (GLM) with robust standard errors to account for the geographical clustering of WASH Benefits households and for repeated measures within the same individual. For each outcome we investigated, we identified potential confounders as factors that are predictive of the dependent variable and also likely to be associated with the independent variables of interest. We considered the following covariates as potential confounders: age of index child, education of mother/caregiver, education of father, household wealth index based on principle components analysis [31, 32], and season. In the multivariable model we included all covariates that were associated with the dependent variable at the p<0.2 level in bivariate analyses.

We also assessed if the associations between child feces management and our study outcomes vary by season. We pre-specified three distinct seasons before examining outcomes: a hot, humid summer (mid-March to mid-June), a cool, rainy monsoon season (mid-June to mid-October) and a cool, dry winter (mid-October to mid-March) to reflect the typical temperature and rainfall patterns of the region [33]. Bangladesh receives over >80% of its rainfall during the monsoon season [34]. Summers and winters are dry, with summer temperatures ranging between 30–40˚C and winter temperatures ranging between 10–30˚C [33]. For each outcome we investigated, we assessed effect modification by season by including interaction terms between the exposure variable and season in the models. We examined the statistical significance of the interaction terms with a Wald test comparing the models with and without the interaction terms, and we interpreted a p-value <0.2 as evidence of effect modification by season.

### Ethical considerations

All households provided written informed consent. The study protocol was reviewed and approved by human subjects review committees at icddr,b (PR-11063), University of California, Berkeley (2011-09-3652), and Stanford University (25863).

## Results

### Household characteristics

Among the 360 households enrolled in this study, mean age of the children at enrollment was 13 months (SD = 2.9). Mean age of the mothers was 24 years (SD = 5), and about 43% of mothers had secondary and above education. The mean number of children <5 years in the compound was 2 (SD = 1.1). About 34% of the households had natural walls (jute, bamboo or mud), 57% of households had electricity and about 86% of households had a cell phone (Table 1).

The most frequently observed drinking water storage containers were pitchers (55%) and *kalash* (a lidless aluminum vessel with a narrow mouth but a wide brim that is typically covered using a plate) (38%). Among these, 81% of pitchers and 77% of *kalash* were observed to be uncovered (Table 2). About 98% of the households had access to a latrine and 64% of households had an improved primary latrine (Table 2). Among the 360 households visited eight times over the study period, there were 2655 reported last child defecation and 2611 reported last child feces disposal events. Among these, 21% (n = 548) reported safe defecation and 23%

**Table 1. Enrollment characteristics of study households with at least one child <5 years in rural Bangladesh (N = 360).**

| Characteristics | | % (n) |
|---|---|---|
| Child age at enrollment in months, mean (SD) | | 13 (2.9) |
| Sex of enrolled child | | |
| | Male | 51 (184) |
| | Female | 49 (176) |
| Number of children <5 yrs in the household, mean (SD) | | 1.3 (0.5) |
| Number of children <5 yrs in the compound [a], mean (SD) | | 2 (1.1) |
| Mother's age in years, mean (SD) | | 24 (5) |
| Mother's education | | |
| | No or primary education | 44 (160) |
| | Secondary and above | 56 (200) |
| Father's education | | |
| | No or primary education | 57 (206) |
| | Secondary and above | 43 (154) |
| Number of rooms in household, mean (SD) | | 1.3 (2) |
| Number of households in the compound [a], mean (SD) | | 1.4 (2.4) |
| Households with: | | |
| | Natural wall (made by jute/ bamboo/mud) | 34 (124) |
| | Electricity | 57 (205) |
| | Refrigerator | 10 (35) |
| | Cell phone | 86 (309) |
| | TV (color or black and white) | 32 (113) |

SD: Standard deviation.

[a] **Compound** is a household or a group of households around a central courtyard.

**Table 2. Water, sanitation and hygiene conditions among enrolled households and reported diarrhea for children <5 years[a].**

| Characteristics | | N | % (n) or mean (SD) |
|---|---|---|---|
| **Water quality indicators** | | | |
| Primary drinking water storage container and covering status | | | |
| Kalash (narrow-mouth container) | | 2353 | 38 (892) |
| | Covered kalash | 892 | 23 (207) |
| | Uncovered kalash | 892 | 77 (685) |
| Pitcher (wide-mouth container) | | 2353 | 55 (1285) |
| | Covered pitcher | 1285 | 19 (245) |
| | Uncovered pitcher | 1285 | 81 (1040) |
| Household reports treating drinking water | | 2353 | 0.4 (10) |
| **Hand hygiene indicators** | | | |
| Observed mother washing hands with only water before collecting hand rinse | | 2656 | 18 (486) |
| Observed mother washing hands with water and soap before collecting hand rinse | | 2656 | 3.1 (81) |
| Observed child washing hands with only water before collecting hand rinse | | 2656 | 4.4 (118) |
| Observed child washing hands with water and soap before collecting hand rinse | | 2656 | 0.7 (19) |
| **Sanitation indicators** | | | |
| Household has access to latrine | | 2656 | 98 (2619) |
| Household has improved primary latrine [b] | | 2605 | 64 (1658) |
| Household has hygienic primary latrine [c] | | 2605 | 38 (983) |
| **Child feces management indicators** | | | |
| Reported safe child defecation for last defecation event [d] | | 2655 | 21 (548) |
| Reported safe child feces disposal practices for last defecation event [e] | | 2611 | 23 (607) |
| Observed human feces within the compound | | 2623 | 0.8 (21) |
| **Visible dirt on caregiver and child hands** | | | |
| Dirty caregiver hands [f] | | 2662 | 79 (2083) |
| Dirty child hands [f] | | 2616 | 67 (1775) |
| **Caregiver reported diarrhea for children <5 years** | | | |
| 2-day prevalence | | 2595 | 7.1 (184) |
| 7-day prevalence | | 2595 | 11.8 (305) |

[a]Using pooled data from all follow-up visits (total 8 visits).

[b] Defined using WHO/UNICEF Joint Monitoring Programme definition for **improved latrine**.

[c] **Hygienic latrine**s include flush/pit latrines with water seal and no visible feces on slab or floor inside and not directly open to the environment.

[d] **Safe child defecation** defined as defecation in a potty or latrine.

[e] **Safe child feces disposal** defined as feces put/rinsed into latrine or specific pit or buried.

[f] **Dirty hands** defined as visible dirt particles on palms, pads or nails of one or both hands.

(n = 607) reported safe feces disposal. Fewer than 1% of households (n = 21) had human feces observed in the compound area. The caregiver-reported prevalence of diarrhea among children <5 years was 7.1% for a 2-day recall window and 11.8% for a 7-day recall window (Table 2).

**Table 3. Presence and concentration of *E. coli* in caregiver and child hand rinse and stored drinking water samples among households with child <5 years in rural Bangladesh.**

| Sample Type | N | Positive % (n) | Geometric mean (GSD) [a] |
|---|---|---|---|
| Caregiver hands | 2662 | 75 (1988) | 1.15 (0.92) |
| Child hands | 2623 | 75 (1963) | 1.17 (0.91) |
| Stored drinking water | 2319 | 81 (1870) | 1.34 (0.93) |

GSD: Geometric Standard Deviation.

[a]log10 MPN per 2 hands for hand rinse samples and per 100 mL for water samples.

## Hand and drinking water contamination

A total of 2662 caregivers hand rinse samples, 2623 child hand rinse samples and 2319 stored drinking water samples were collected from 360 household over eight visits. Among these, 75% (n = 1988) of caregiver hand rinse samples, 75% (n = 1963) of child hand rinse samples and 81% (n = 1870) of stored drinking water samples were *E. coli* positive. The geometric mean *E. coli* count on caregiver and child hands was 1.15 (SD = 0.92) and 1.17 (SD = 0.91) $\log_{10}$ MPN per 2 hands, respectively, and 1.34 (SD = 0.93) per 100 mL for stored drinking water (Table 3). We observed visible dirt on 67% (n = 1775) of child hands and 79% (n = 2083) of caregiver hands (Table 2).

## Unadjusted analyses

Prevalence of *E. coli* in stored drinking water and caregiver and child hand rinse samples was significantly higher among households where unsafe (vs. safe) defecation and unsafe (vs. safe) child feces disposal was reported for children <5 years (Table 4). Levels of *E. coli* in child hand rinse samples were significantly higher among households with unsafe child defecation (Table 5). Levels of *E. coli* in stored water samples were significantly higher among households

**Table 4. Reported child defecation and child feces disposal practices vs. prevalence of *E. coli* on caregiver and child hands and in stored drinking water.**

| | | *E. coli* in caregiver hand rinse samples, N = 1988 | | | | *E. coli* in child hand rinse samples, N = 1963 | | | | *E. coli* in stored drinking water samples, N = 1870 | | | |
|---|---|---|---|---|---|---|---|---|---|---|---|---|---|
| | | N | Positive | Bivariate model[a] | Multivariable model[b] | N | Positive | Bivariate model[a] | Multivariable model[b] | N | Positive | Bivariate model[a] | Multivariable model[b] |
| | | | % (n) | PR (95% CI) | PR (95% CI) | | % (n) | PR (95% CI) | PR (95% CI) | | % (n) | PR (95% CI) | PR (95% CI) |
| Last reported child defecation | | | | | | | | | | | | | |
| | Safe | 574 | 68 (395) | Ref | Ref | 593 | 67 (400) | Ref | Ref | 495 | 72 (357) | Ref | Ref |
| | Unsafe | 2088 | 76 (1593) | **1.11 (1.02, 1.20)** | 1.05 (0.97, 1.14) | 2030 | 77 (1563) | **1.14 (1.07, 1.22)** | **1.12 (1.04, 1.20)** | 1824 | 83 (1513) | **1.15 (1.07, 1.24)** | **1.12 (1.03, 1.21)** |
| Last reported child feces disposal | | | | | | | | | | | | | |
| | Safe | 607 | 70 (426) | Ref | Ref | 607 | 7 (409) | Ref | Ref | 498 | 73 (366) | Ref | Ref |
| | Unsafe | 2055 | 76 (1562) | **1.08 (1.01, 1.16)** | 1.03 (0.97, 1.10) | 2016 | 77 (1554) | **1.14 (1.06, 1.24)** | **1.11 (1.02, 1.21)** | 1821 | 83 (1504) | **1.13 (1.06, 1.20)** | **1.10 (1.03, 1.18)** |

PR: Prevalence ratio; CI: Confidence interval.

[a]We estimated the prevalence ratio by using generalized linear models (GLM) with robust standard errors to adjust for clustering within study clusters and within repeated measurements from the same household.

[b] Multivariable model includes all variables associated with *E. coli* in samples in bivariate analyses at p<0.2 level.

**Table 5. Reported child defecation and child feces disposal practices vs. level of *E. coli* on caregiver and child hands and in stored drinking water.**

| | | *E. coli* in caregiver hand rinse samples, N = 1988 | | | | *E. coli* in child hand rinse samples, N = 1963 | | | | *E. coli* in stored drinking water samples, N = 1870 | | | |
|---|---|---|---|---|---|---|---|---|---|---|---|---|---|
| | | N | Log₁₀ mean MPN (SD) | Bivariate model[a] | Multivariable model[b] | N | Log₁₀ mean MPN (SD) | Bivariate model[a] | Multivariable model[b] | N | Log₁₀ mean MPN (SD) | Bivariate model[a] | Multivariable model[b] |
| | | | | $\Delta$log10 (95% CI) | $\Delta$log10 (95% CI) | | | $\Delta$log10 (95% CI) | $\Delta$log10 (95% CI) | | | $\Delta$log10 (95% CI) | $\Delta$log10 (95% CI) |
| Last reported child defecation | | | | | | | | | | | | | |
| | Safe | 395 | 1.08 (0.91) | Ref | Ref | 400 | 1.06 (0.92) | Ref | Ref | 357 | 1.19 (0.91) | Ref | Ref |
| | Unsafe | 1593 | 1.16 (0.92) | 0.08 (-0.03, 0.20) | 0.04 (-0.07, 0.16) | 1563 | 1.20 (0.90) | **0.14 (0.03, 0.25)** | 0.07 (-0.04, 0.19) | 1513 | 1.37 (0.93) | **0.18 (0.06 0.30)** | **0.15 (0.03, 0.27)** |
| Last reported child feces disposal | | | | | | | | | | | | | |
| | Safe | 426 | 1.07 (0.91) | Ref | Ref | 409 | 1.10 (0.92) | Ref | Ref | 366 | 1.21 (0.94) | Ref | Ref |
| | Unsafe | 1562 | 1.17 (0.92) | 0.10 (-0.02, 0.21) | 0.06 (-0.05, 0.17) | 1554 | 1.19 (0.90) | 0.09 (-0.02, 0.20) | 0.02 (-0.09, 0.14) | 1504 | 1.37 (0.92) | **0.16 (0.04, 0.27)** | **0.11 (0.01, 0.23)** |

MPN: Most probable number; CI: Confidence interval; SD: Standard deviation.

[a]We estimated the log₁₀ difference by using generalized linear models (GLM) with robust standard errors to adjust for clustering within study clusters and within repeated measurements from the same household.

[b] Multivariable model includes all variables associated with *E. coli* in samples in bivariate analyses at p<0.2 level.

with unsafe child defecation and feces disposal (Table 5). There was no association between *E. coli* levels in caregiver hand rinse samples and reported child defecation child feces disposal practices (Table 5). In households where unsafe child defecation and feces disposal was reported, children were more likely to have visible dirt on their hands but there was also no statistically significant association between the prevalence of visible dirt on caregiver hands and reported child defecation or feces disposal practices (Table 6). Children in households

**Table 6. Reported child defecation and child feces disposal practices vs. observed cleanliness of caregiver and child hands.**

| | | Dirty caregiver hands[a] (N = 1767) | | | | Dirty child hands[a] (N = 2077) | | | |
|---|---|---|---|---|---|---|---|---|---|
| | | N | % (n) | Bivariate model[b] PR (95% CI) | Multivariable model[c] PR (95% CI) | N | % (n) | Bivariate model[b] PR (95% CI) | Multivariable model[c] PR (95% CI) |
| Last reported child defecation | | | | | | | | | |
| | Safe | 574 | 66 (377) | Ref | Ref | 593 | 67 (401) | Ref | Ref |
| | Unsafe | 2078 | 67 (1390) | 1.02 (0.90, 1.16) | 1.01 (0.89, 1.14) | 2023 | 82 (1676) | **1.23 (1.08, 1.39)** | **1.18 (1.04, 1.33)** |
| Last reported child feces disposal | | | | | | | | | |
| | Safe | 607 | 61 (371) | Ref | Ref | 607 | 70 (427) | Ref | Ref |
| | Unsafe | 2040 | 68 (1396) | 1.12 (0.97, 1.29) | 1.04 (0.90, 1.21) | 2004 | 82 (1650) | **1.18 (1.07, 1.31)** | **1.13 (1.02, 1.25)** |

PR: Prevalence ratio; CI: Confidence interval.

[a] Dirty hands defined as visible dirt on palms, pads or nails of one or both hands.

[b] We estimated the prevalence ratio by using generalized linear models (GLM) with robust standard errors to adjust for clustering within study clusters and within repeated measurements from the same individual.

[c] Multivariable model includes all variables associated with visible dirt on caregiver and child hands in bivariate analyses at p<0.2 level.

**Table 7. Reported child defecation and child feces disposal practices vs. caregiver-reported diarrhea prevalence among children <5 years.**

| | | 2-day prevalence of diarrhea [a] (N = 2595) | | | | 7-day prevalence of diarrhea (N = 2595) | | | |
| | | | | Bivariate model[b] | Multivariable model[c] | | | Bivariate model[b] | Multivariable model[c] |
| | | N | % (n) | PR (95% CI) | PR (95% CI) | N | % (n) | PR (95% CI) | PR (95% CI) |
|---|---|---|---|---|---|---|---|---|---|
| Last reported child defecation | | | | | | | | | |
| | Safe | 578 | 4.8 (28) | Ref | Ref | 578 | 9.0 (52) | Ref | Ref |
| | Unsafe | 2017 | 7.7 (156) | 1.60 (0.72, 3.54) | 1.39 (0.61, 3.16) | 2017 | 12.5 (253) | 1.39 (0.77, 2.52) | 1.20 (0.65, 2.24) |
| Last reported child feces disposal | | | | | | | | | |
| | Safe | 599 | 4.1 (24) | Ref | Ref | 599 | 6.2 (37) | Ref | Ref |
| | Unsafe | 1991 | 8.1 (160) | 2.01 (0.90, 4.49) | 1.69 (0.70, 4.10) | 1991 | 13.5 (268) | **2.18 (1.16, 4.11)** | 1.74 (0.88, 3.44) |

PR: Prevalence ratio; CI: Confidence interval.

[a] Diarrhea defined as three or more loose or watery stools in 24 hours.

[b] We estimated the prevalence ratio by using generalized linear models (GLM) with robust standard errors to adjust for clustering within study clusters and within repeated measurements from the same child.

[c] Multivariable model includes all variables associated with child diarrhea in bivariate analyses at p<0.2 level.

with unsafe child defecation and feces disposal had higher prevalence of diarrhoea measured both with 2-day and 7-day recall but the only statistically significant association was the one between unsafe child feces disposal and 7-day diarrhea prevalence (Table 7).

## Adjusted analyses

In multivariable models controlling for child age, household wealth and mothers' education, the prevalence of *E. coli* in caregiver hand rinse samples was no longer associated with reported child defecation and child feces disposal practices. *E. coli* prevalence in child hand rinse samples remained significantly higher among households reporting unsafe child defecation (PR: 1.12, 1.04–1.20) and unsafe child feces disposal (PR: 1.11, 1.02–1.21). *E. coli* prevalence in stored drinking water was also significantly higher in households reporting unsafe child defecation (PR: 1.12, 1.03–1.21) and unsafe child feces disposal (PR: 1.10, 1.03–1.18) (Table 4). *E. coli* levels on child hand rinse samples were no longer associated with reported child defecation and child feces disposal practices while levels of *E. coli* in stored drinking water remained significantly higher in households reporting unsafe child defecation (Δlog10: 0.15, 0.03–0.27) and unsafe child feces disposal (Δlog10: 0.11, 0.01–0.23) (Table 5). Similarly, the prevalence of visible dirt on child hands remained significantly higher among households with unsafe defecation and feces disposal (Table 6). The magnitude of effect estimates suggested higher 2-day prevalence of child diarrhea in households with unsafe defecation (PR: 1.39, 0.61–3.16) and unsafe feces disposal (PR: 1.69, 0.70–4.10) but these associations remained statistically non-significant (Table 7). Effects were similar for 7-day prevalence of diarrhea (Table 7).

## Effect modification by season

The prevalence of *E. coli* on caregiver hands was 69% in the summer, 76% during the monsoon and 77% in the winter, while the prevalence of *E. coli* on child hands was 65% in the summer, 77% during the monsoon and 79% in the winter. The prevalence of *E. coli* in stored drinking water samples in the summer, monsoon and winter seasons was 69%, 76% and 77%, respectively (S1 Table). The prevalence of visible dirt on caregiver hands was similar (66–67%) across the seasons as was the prevalence of visible dirt on child hands (78–80%) (S1 Table). The caregiver-reported 2-day prevalence of diarrhea among children <5 years was 8.1% in the summer,

6.8% in the monsoon and 7.9% in the winter, while the 7-day prevalence of diarrhoea was 12% in the summer, 11% in the monsoon and 13% in the winter (S1 Table).

Subgroup analyses suggested that the association between unsafe child feces management and *E. coli* contamination of caregiver hands and stored water was more pronounced during the summer season than during the monsoon or winter seasons (interaction p-values <0.05) (S2 Table).

## Discussion

The nationwide estimate for open defecation, as defined by lack of latrine access, is 2% in Bangladesh [2]. In our study, 98% of households had access to a latrine, consistent with these estimates. However, the majority of households reported unsafe child defecation and unsafe disposal of child feces, suggesting that open defecation by young children is common in this setting despite widespread access to on-site sanitation. Our findings are consistent with other studies in rural Bangladesh that found 74% unsafe child defecation and 80% unsafe child feces disposal reported by caregivers [9, 35], as well as three studies in India [23, 24, 36] and one study in Ethiopia reporting unsafe child defecation in 54–80% and unsafe child feces disposal in 67–81% of households [37]. Taken together, these studies suggest that, among households with young children, three quarters could be at risk of pathogen exposure from child feces in the home environment even when a latrine is present.

Our findings of increased fecal contamination associated with unsafe child feces management are consistent with evidence from other settings. A study in India found that *E. coli* counts on household floors and in soil increased by up to an order of magnitude following child defecation on these surfaces after the feces were removed [23]. A study in Kenya using microbial source tracking methods to distinguish the feces of young children from the feces of older children and adults found that fecal contamination from young children dominated samples collected within the domestic environment, such as hands and surfaces [38].

We did not find an association between reported child defecation or feces disposal practices and *E. coli* contamination or visual cleanliness of caregiver hands, while child hands in households with unsafe child defecation and feces disposal were more likely to be contaminated by *E. coli* and be visibly dirty. One possible explanation for the lack of association between contamination of mothers' hands and child feces management could be that *E. coli* levels on caregiver hands are highly temporally variable and fluctuate in response to various domestic tasks, which could mask any effect of exposure to child feces [39]. A study in India found an increase in *E. coli* counts on hands of caregivers after they handled child feces with unsafe methods but not with safe methods [23]. This study measured caregiver hand contamination immediately following feces handling events while we collected hand rinses at an arbitrary time during the interview. Our sampling method likely missed spikes in caregiver hand contamination associated with unsafe feces handling due to temporal variability. In contrast, our findings suggest that open child feces in the domestic environment increase the risk of fecal exposure among young children through contaminated hands. This could be because children spend time exploring the home environment and have frequent hand contact with feces or with soil contaminated by feces. Children's interactions with the environment increase their risk of exposure to highly contaminated reservoirs like soil contaminated with lead [40], pesticides in agricultural communities [41], arsenic in water [42] or animal feces or animal manure used as cooking fuel [43]. A study in rural Bangladesh found that, in 5% of eating events, children's hands contacted soil that may be highly contaminated by feces [15]. A study in Tanzania found that children placing contaminated hands in their mouths accounted for 97% of the

total quantity of ingested fecal matter whereas only 3% was due to direct consumption of contaminated drinking water [44].

There is mixed evidence on the effect of overall sanitation improvements on hand cleanliness. A systematic review found that sanitation programs did not reduce fecal contamination on most transmission pathways including hands [10]. An observational study in Tanzania showed that improved sanitation was associated with reduction of fecal indicator bacteria on mothers' hands [45] whereas a school-based randomized controlled trial in Kenya found that provision of latrines was associated with increased hand contamination among students [46], suggesting child hand contamination may be insensitive to sanitation improvements without accompanying improvements in hygiene. It is possible that sanitation programs, which typically focus on the feces of adults and older children, are insufficient to reduce fecal contamination in the home environment without measures for hygienic defecation and feces disposal for young children. The WASH Benefits Bangladesh trial, whose control arm for this study was nested in, had a sanitation intervention arm that received latrine upgrades, as well as child potties and scoops for removal of child and animal feces. The trial found borderline reductions in visible cleanliness of caregiver finger pads and palms, and no reductions in visible cleanliness or *E. coli* contamination of child hands among participants of the sanitation arm compared to control participants receiving no intervention [9]. This could have occurred because child feces management practices remained poor among recipients of the sanitation intervention despite access to hardware; respondents reported that <20% of children defecated safely in the latrine/potty, <30% of households disposed of child feces in the latrine and <10% of households used the scoop to handle child feces [9].

Our findings also suggest higher levels of *E. coli* contamination in stored drinking water in households with unsafe child feces management. Our study was not designed to differentiate contamination occurring at the source from contamination introduced during storage at home. The increased contamination we observed could be due to child feces in the compound environment entering the tubewell by subsurface infiltration or through unsealed head works [47, 48], or due to contact with hands and utensils during storage that have been contaminated by exposure to child feces [49, 50]. Several previous randomized control trials that assessed the association between sanitation and drinking water quality found no effect from sanitation [5–7], while a trial in Tanzania found reduced *E. coli* in drinking water associated with sanitation improvements [51]. In addition, several observational studies found no association between sanitation and drinking water quality [26, 52, 53], while an observational study in Indonesia found reduced fecal contamination of drinking water associated with improved sanitation [54].

Our findings suggest that unsafe child defecation and unsafe child feces disposal are associated with increased *E. coli* contamination of child hands and stored drinking water, suggesting the possibility of an increased risk of child gastrointestinal illness. The evidence to date on the association between child feces management and child diarrhea is mixed. Two randomized controlled trials conducted in rural Bangladesh found 27–30% reduction in pediatric diarrhea associated with disposing of child feces in a latrine and no visible feces being present in the household compound [55, 56]. In addition, unsafe child defecation and feces disposal behaviors were found to be associated with an increased risk of diarrheal diseases in an observational study conducted in Indonesia [57]. In contrast, a study in rural Bangladesh did not find any association between unsafe defecation and unsafe feces disposal and child gastrointestinal illness [35]. A recent meta-analysis that assessed the health impact of safe defecation and safe feces disposal showed that, out of five studies reviewed, only two found a reduction in diarrhea while the others did not find an association [58]. A study of Demographic and Health Survey (DHS) data from 34 countries also showed improved child growth associated with safe

disposal of child feces [22]. The magnitude of effect estimates in our study suggested higher risk of diarrhea associated with unsafe child feces management; however, we could not rule out chance as the explanation for these associations. This could be because the analysis using diarrhea as the outcome had lower statistical power than those focused on the *E. coli* outcomes as the prevalence of diarrhea was low compared to the prevalence of *E. coli* in our samples.

## Limitations

Our study had some limitations in terms of exposure and outcome measurements. We recorded caregiver-reported child defecation and child feces management practices which could be subject to courtesy bias and underestimate true levels by underreporting socially undesirable behaviors. Further, *E. coli* measurements do not distinguish between human vs. animal fecal sources [59] or between *E. coli* of fecal vs. natural origin [60]; this complicates interpretation of *E. coli* contamination detected along different pathways. Additionally, we relied on caregiver-reported diarrhea symptoms which can have inaccurate recall and also do not capture asymptomatic sub-clinical infections, which are common in low-income country settings [61].

Our analysis was observational and therefore susceptible to confounding. While we controlled for potential confounding factors in our analysis, it is possible that residual confounding remains from unmeasured factors. For example, households that practice unsafe child feces management could also have poorer hand hygiene and water handling practices. However, we did not find an association between child feces management and contamination of caregiver hands, suggesting no blanket confounding in our findings from unmeasured factors.

## Conclusion

Unsafe child defecation and child feces disposal was reported by the majority of households in a rural Bangladeshi setting with widespread access to on-site sanitation. These practices were associated with increased *E. coli* contamination of child hands and stored drinking water, increased likelihood of visible dirt on child hands, and potentially increased risk of diarrhea. Our findings suggest that child open defecation and poor child feces management may be sources of fecal exposure for young children. Studies should assess if targeted interventions to improve child feces management practices reduce fecal contamination in the domestic environment and child diarrhea.

## Supporting information

**S1 Table. Mean *E. coli* prevalence and log$_{10}$ MPN count by season (summer vs. monsoon vs. winter season).**
(DOCX)

**S2 Table. Subgroup analysis by summer vs. monsoon/wet vs. winter/dry season on *E. coli* prevalence.**
(DOCX)

## Acknowledgments

The authors would like to thank the participants for donating their time and the field team for collecting the data. We also thank all the collaborators engaged in the study.

## Author Contributions

**Conceptualization:** Mahfuza Islam, Mahbubur Rahman, Leanne Unicomb, Amy J. Pickering, Stephen P. Luby, Benjamin F. Arnold, John M. Colford, Jr., Ayse Ercumen.

**Data curation:** Mahfuza Islam, Mohammad Abdullah Heel Kafi, Mostafizur Rahman, Mahfuja Alam, Ayse Ercumen.

**Formal analysis:** Mahfuza Islam, Leanne Unicomb, Mohammad Abdullah Heel Kafi, Mostafizur Rahman, Debashis Sen, Sharmin Islam, Amy J. Pickering, Alan E. Hubbard, Stephen P. Luby, Benjamin F. Arnold, John M. Colford, Jr., Ayse Ercumen.

**Funding acquisition:** Ayse Ercumen.

**Investigation:** Mahfuza Islam, Ayse Ercumen.

**Methodology:** Mahfuza Islam, Mahbubur Rahman, Leanne Unicomb, Mostafizur Rahman, Mahfuja Alam, Debashis Sen, Ayse Ercumen.

**Project administration:** Mahfuza Islam, Mahbubur Rahman, Mostafizur Rahman, Mahfuja Alam, Debashis Sen, Sharmin Islam, Stephen P. Luby, John M. Colford, Jr., Ayse Ercumen.

**Supervision:** Mahfuza Islam, Mahbubur Rahman, Mostafizur Rahman, Mahfuja Alam, Debashis Sen, Sharmin Islam, Ayse Ercumen.

**Validation:** Mahfuza Islam, Mahbubur Rahman, Mohammad Abdullah Heel Kafi, Mostafizur Rahman, Debashis Sen, Sharmin Islam, Amy J. Pickering, Alan E. Hubbard, Stephen P. Luby, Benjamin F. Arnold, John M. Colford, Jr., Ayse Ercumen.

**Visualization:** Mahfuza Islam, Mahbubur Rahman, Mohammad Abdullah Heel Kafi, Mostafizur Rahman, Debashis Sen, Sharmin Islam, Amy J. Pickering, Alan E. Hubbard, Stephen P. Luby, Benjamin F. Arnold, John M. Colford, Jr., Ayse Ercumen.

**Writing – original draft:** Mahfuza Islam, Mahbubur Rahman, Leanne Unicomb, Mohammad Abdullah Heel Kafi, Mostafizur Rahman, Mahfuja Alam, Debashis Sen, Sharmin Islam, Amy J. Pickering, Alan E. Hubbard, Stephen P. Luby, Benjamin F. Arnold, John M. Colford, Jr., Ayse Ercumen.

**Writing – review & editing:** Mahfuza Islam, Mahbubur Rahman, Leanne Unicomb, Alan E. Hubbard, Stephen P. Luby, Benjamin F. Arnold, John M. Colford, Jr., Ayse Ercumen.

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
