## [Decision Letter · Decision Letter 0]

1 Apr 2020

PONE-D-20-03333

Child defecation and child feces management practices in rural Bangladeshi households: Associations with fecal contamination of hands and stored drinking water, observed hand cleanliness and child diarrhea

PLOS ONE

Dear Mrs Islam,

Thank you for submitting your manuscript to PLOS ONE. After careful consideration, we feel that it has merit but does not fully meet PLOS ONE’s publication criteria as it currently stands. Therefore, we invite you to submit a revised version of the manuscript that addresses the points raised during the review process.

We would appreciate receiving your revised manuscript by May 16 2020 11:59PM. To enhance the reproducibility of your results, we recommend that if applicable you deposit your laboratory protocols in protocols.io, where a protocol can be assigned its own identifier (DOI) such that it can be cited independently in the future. For instructions see: http://journals.plos.org/plosone/s/submission-guidelines#loc-laboratory-protocols

We look forward to receiving your revised manuscript.

Kind regards,

William Joe

Academic Editor

PLOS ONE

Journal Requirements:

'This study was funded by National Institute 435 of Health (NIH), grant number R01HD078912.'

'The funders had no role in study design, data collection and analysis, decision to publish, or preparation of the manuscript.'

Additional Editor Comments (if provided):

Reviewers' comments:

Reviewer's Responses to Questions

**Comments to the Author**

1. Is the manuscript technically sound, and do the data support the conclusions?

Reviewer #1: Yes

Reviewer #2: Yes

2. Has the statistical analysis been performed appropriately and rigorously? 

Reviewer #1: Yes

Reviewer #2: Yes

3. Have the authors made all data underlying the findings in their manuscript fully available?

Reviewer #1: Yes

Reviewer #2: Yes

4. Is the manuscript presented in an intelligible fashion and written in standard English?

Reviewer #1: Yes

Reviewer #2: Yes

5. Review Comments to the Author

Reviewer #1: Overall this is a good study but needs to be reviewed and toned down by the authors for the following checks:

1. Title/s: The authors may revise the language of the title/s to improve readability

2. The review of literature needs to be more critically written: from global statistics, it can be narrowed down to Bangladesh, and more data should be provided particularly in terms of low- and middle-income countries. The gap in literature needs to be highlighted more. Line 52 and 53 – repetition. Overall, the literature review section needs to be worked upon, especially the language to improve readability

3. In the study design section of methodology, line 78 – Instead of ‘longitudinal environmental assessment’, can be mentioned only as Longitudinal study. The study design needs explicit description.

4. In the results section, line number 205 – ‘Human faeces were observed in <1% (n=21) of households’: The sentence is not clear, may be reframed.

5. In Table 1: Enrolment characteristics of study households with at least one child <5 years in rural Bangladesh (N=360), what does ‘Female” in child characteristics denote? Is it percentage of female child in the house? If yes, then why just female and not the male child percentage? – This table needs to be reworked, the variables under each head needs to be reviewed. For example, under ‘Child characteristics’, there is ‘mother’s age in years’, which should go under a different head.

Reviewer #2: The manuscript is technically sound it tends to explain the objective of the study and how they have selected cluster for the observation of sanitation practices. It is aptly written, however i will suggest that they should mention gaps in the existing literature in the initial part of the paper. Further the writing part in the study design, sample collection, and analysis can be simplified for the wider audience for instance using the term index children can be explained and why they have used that term. Over all the paper present clear picture of sanitation practices in Bangladesh.

6. PLOS authors have the option to publish the peer review history of their article (what does this mean?). If published, this will include your full peer review and any attached files.

Reviewer #1: Yes: Poulami Dasgupta

Reviewer #2: No

---

## [Author Response · Author response to Decision Letter 0]

21 Apr 2020

Editor

PLOS ONE

Thank you for your helpful comments and suggestions. Please find attached our responses to reviewers’ comments. Under each response, we have also included the relevant excerpt from the revised manuscript to facilitate reviewing. 

Thank you for your consideration, and we look forward to your decision.

Sincerely,

Mahfuza Islam

JOURNAL REQUIREMENTS:

1. Please ensure that your manuscript meets PLOS ONE's style requirements

Response: We have followed PLOS ONE’s required style including formatting and file naming.

'This study was funded by National Institute 435 of Health (NIH), grant number R01HD078912.'

'The funders had no role in study design, data collection and analysis, decision to publish, or preparation of the manuscript.'

Response: We have removed the funding related text from the “Acknowledgement” section of the manuscript. Please revise our Funding Statement in the online system as follows.

'This study was funded by National Institutes of Health (NIH) grant number R01HD078912.The funder had no role in study design, data collection and analysis, decision to publish, or preparation of the manuscript.'

RESPONSES TO REVIEWERS:

RESPONSES TO REVIEWER #1:

Overall this is a good study but needs to be reviewed and toned down by the authors for the following checks:

1. Title/s: The authors may revise the language of the title/s to improve readability

Response: We have revised the study title as follows.

Manuscript Revision:

“Child defecation and feces management practices in rural Bangladesh: Associations with fecal contamination, observed hand cleanliness and child diarrhea”

2. The review of literature needs to be more critically written: from global statistics, it can be narrowed down to Bangladesh, and more data should be provided particularly in terms of low- and middle-income countries. The gap in literature needs to be highlighted more. Line 52 and 53 – repetition. Overall, the literature review section needs to be worked upon, especially the language to improve readability

Response: We have revised the sentence (line 52 and 53) as follows. We have also expanded the literature review in the introduction and discussion sections and improved the language.

Manuscript Revision:

“…Young children frequently place their hands in their mouths, and in Bangladesh, it is also common to eat and to be fed by hand (8). Previous studies in Bangladesh demonstrated that caregiver’ and children’s hands can contain fecal indicator organisms at concentrations of >100 colony forming units (CFU) per two hands (9)...”

Expanded literature in the introduction section

“…However, several recent sanitation trials have shown mixed impact from latrine provision on health outcomes (3-7) and studies that measured fecal contamination at potential household exposure points found little or no effect of sanitation interventions in reducing fecal indicator bacteria (8-10), suggesting other sources of fecal contamination that are not adequately eliminated by typical sanitation interventions. One potential source is child open defecation, which remains common in low-income countries....”

“…Poor child feces management could be a potential contributor to health risk as young children with poorly developed immune systems have higher incidence of enteric infections than other age groups (12) and their feces are also more likely to contain higher quantities of transmissible pathogens (13)....”

Expanded literature in the discussion section

 “…Our findings of increased fecal contamination associated with unsafe child feces management are consistent with evidence from other settings. A study in India found that E. coli counts on household floors and in soil increased by up to an order of magnitude following child defecation on these surfaces after the feces were removed (33). A study in Kenya using microbial source tracking methods to distinguish the feces of young children from the feces of older children and adults found that fecal contamination from young children dominated samples collected within the domestic environment, such as hands and surfaces (35)...”

“…A study in India found an increase in E. coli counts on hands of caregivers after they handled child feces with unsafe methods but not with safe methods (33). This study measured caregiver hand contamination immediately following feces handling events while we collected hand rinses at an arbitrary time during the interview. Our sampling method likely missed spikes in caregiver hand contamination associated with unsafe feces handling due to temporal variability...”

“…A study of Demographic and Health Survey (DHS) data from 34 countries also showed improved child growth associated with safe disposal of child feces (57)...”

3. In the study design section of methodology, line 78 – Instead of ‘longitudinal environmental assessment’, can be mentioned only as Longitudinal study. The study design needs explicit description.

Response: We have added additional details of the study design and changed the wording to longitudinal study. We have also added citations for previous publications that describe additional details of the study design and implementation.

Manuscript Revision:

“…We conducted a longitudinal study within a randomized controlled trial in rural Bangladesh (WASH Benefits Bangladesh trial, ClinicalTrials.gov NCT01590095). The parent trial was conducted in the Gazipur, Kishoreganj, Mymensingh and Tangail districts of rural Bangladesh (18, 19). The trial randomly assigned geographically pair-matched clusters of pregnant women to water, sanitation, hygiene and nutrition intervention vs. control arms and followed their birth cohort of “index children” (children of enrolled pregnant women were in utero at the time of enrollment) for approximately two years to assess intervention impacts on child growth, diarrhea and enteric infections. Additional details of the study design and interventions have been described elsewhere (18-20)...”

4. In the results section, line number 205 – ‘Human faeces were observed in <1% (n=21) of households’: The sentence is not clear, may be reframed.

Response: We have revised the sentence as follows.

Manuscript Revision:

“…Fewer than 1% of households (n=21) had human feces observed in the compound area...”

5. In Table 1: Enrolment characteristics of study households with at least one child <5 years in rural Bangladesh (N=360), what does ‘Female” in child characteristics denote? Is it percentage of female child in the house? If yes, then why just female and not the male child percentage? – This table needs to be reworked, the variables under each head needs to be reviewed. For example, under ‘Child characteristics’, there is ‘mother’s age in years’, which should go under a different head.

Response: Yes, it is the percentage of female children of the households which was 49%, and the rest of the 51% were male children. We have added the percentage of the male children to the table.

We have also changed the heading to “Characteristics” instead of “Child characteristics”. 

RESPONSES TO REVIEWER #2: 

Reviewer #2: The manuscript is technically sound it tends to explain the objective of the study and how they have selected cluster for the observation of sanitation practices. It is aptly written, however i will suggest that they should mention gaps in the existing literature in the initial part of the paper. Further the writing part in the study design, sample collection, and analysis can be simplified for the wider audience for instance using the term index children can be explained and why they have used that term. Over all the paper present clear picture of sanitation practices in Bangladesh.

Response: Thanks for your positive comments. We have simplified the language throughout the manuscript and we have added the definition of “index children” as follows. We have also expanded the literature review in the introduction and discussion sections.

Manuscript Revision:

“…The trial randomly assigned geographically pair-matched clusters of pregnant women to water, sanitation, hygiene and nutrition intervention vs. control arms and followed their birth cohort of “index children” (children of enrolled pregnant women that were in utero at the time of enrollment) for approximately two years to assess intervention impacts on child growth, diarrhea and enteric infections. Additional details of the study design and interventions have been described elsewhere (18-20)...”

Expanded literature in the introduction section

“…However, several recent sanitation trials have shown mixed impact from latrine provision on health outcomes (3-7) and studies that measured fecal contamination at potential household exposure points found little or no effect of sanitation interventions in reducing fecal indicator bacteria (8-10), suggesting other sources of fecal contamination that are not adequately eliminated by typical sanitation interventions. One potential source is child open defecation, which remains common in low-income countries....”

“…Poor child feces management could be a potential contributor to health risk as young children with poorly developed immune systems have higher incidence of enteric infections than other age groups (12) and their feces are also more likely contain higher quantities of transmissible pathogens (13)....”

Expanded literature in the discussion section

 “…Our findings of increased fecal contamination associated with unsafe child feces management are consistent with evidence from other settings. A study in India found that E. coli counts on household floors and in soil increased by up to an order of magnitude following child defecation on these surfaces after the feces were removed (33). A study in Kenya using microbial source tracking methods to distinguish the feces of young children from the feces of older children and adults found that fecal contamination from young children dominated samples collected within the domestic environment, such as hands and surfaces (35)...”

“…A study in India found an increase in E. coli counts on hands of caregivers after they handled child feces with unsafe methods but not with safe methods (33). This study measured caregiver hand contamination immediately following feces handling events while we collected hand rinses at an arbitrary time during the interview. Our sampling method likely missed spikes in caregiver hand contamination associated with unsafe feces handling due to temporal variability...”

“…A study of Demographic and Health Survey (DHS) data from 34 countries also showed improved child growth associated with safe disposal of child feces (57)...”

---

## [Decision Letter · Decision Letter 1]

4 Jun 2020

PONE-D-20-03333R1

Child defecation and feces management practices in rural Bangladesh: Associations with fecal contamination, observed hand cleanliness and child diarrhea

PLOS ONE

Dear Dr. Islam,

Thank you for submitting your manuscript to PLOS ONE. After careful consideration, we feel that it has merit but does not fully meet PLOS ONE’s publication criteria as it currently stands. Therefore, we invite you to submit a revised version of the manuscript that addresses the points raised during the review process.

We look forward to receiving your revised manuscript.

Kind regards,

William Joe

Academic Editor

PLOS ONE

Reviewers' comments:

Reviewer's Responses to Questions

**Comments to the Author**

1. If the authors have adequately addressed your comments raised in a previous round of review and you feel that this manuscript is now acceptable for publication, you may indicate that here to bypass the “Comments to the Author” section, enter your conflict of interest statement in the “Confidential to Editor” section, and submit your "Accept" recommendation.

Reviewer #1: All comments have been addressed

Reviewer #2: All comments have been addressed

2. Is the manuscript technically sound, and do the data support the conclusions?

Reviewer #1: Yes

Reviewer #2: Yes

3. Has the statistical analysis been performed appropriately and rigorously? 

Reviewer #1: Yes

Reviewer #2: I Don't Know

4. Have the authors made all data underlying the findings in their manuscript fully available?

Reviewer #1: Yes

Reviewer #2: Yes

5. Is the manuscript presented in an intelligible fashion and written in standard English?

Reviewer #1: Yes

Reviewer #2: Yes

6. Review Comments to the Author

Reviewer #1: I have attached the manuscript with track changes. There are 2 very small changes from my side which can be included. The paper reads really well and is easily understandable.

Reviewer #2: (No Response)

7. PLOS authors have the option to publish the peer review history of their article (what does this mean?). If published, this will include your full peer review and any attached files.

Reviewer #1: Yes: Poulami Dasgupta

Reviewer #2: No

---

## [Author Response · Author response to Decision Letter 1]

11 Jun 2020

I would like to thank both reviewers for giving time to review this manuscript and for providing their valuable comments to improve this.

---

## [Decision Letter · Decision Letter 2]

1 Jul 2020

Child defecation and feces management practices in rural Bangladesh: Associations with fecal contamination, observed hand cleanliness and child diarrhea

PONE-D-20-03333R2

Dear Dr. Islam,

We’re pleased to inform you that your manuscript has been judged scientifically suitable for publication and will be formally accepted for publication once it meets all outstanding technical requirements.

Kind regards,

William Joe

Academic Editor

PLOS ONE

Additional Editor Comments (optional):

Reviewers' comments:

Reviewer's Responses to Questions

**Comments to the Author**

1. If the authors have adequately addressed your comments raised in a previous round of review and you feel that this manuscript is now acceptable for publication, you may indicate that here to bypass the “Comments to the Author” section, enter your conflict of interest statement in the “Confidential to Editor” section, and submit your "Accept" recommendation.

Reviewer #1: All comments have been addressed

Reviewer #2: All comments have been addressed

2. Is the manuscript technically sound, and do the data support the conclusions?

Reviewer #1: Yes

Reviewer #2: Yes

3. Has the statistical analysis been performed appropriately and rigorously? 

Reviewer #1: Yes

Reviewer #2: Yes

4. Have the authors made all data underlying the findings in their manuscript fully available?

Reviewer #1: Yes

Reviewer #2: Yes

5. Is the manuscript presented in an intelligible fashion and written in standard English?

Reviewer #1: Yes

Reviewer #2: Yes

6. Review Comments to the Author

Reviewer #1: The paper really reads well now. No further changes from my side. I wish the authors good luck . Best wishes.

Reviewer #2: (No Response)

7. PLOS authors have the option to publish the peer review history of their article (what does this mean?). If published, this will include your full peer review and any attached files.

Reviewer #1: **Yes: **Poulami Dasgupta

Reviewer #2: No

---

## [Editor Report · Acceptance letter]

8 Jul 2020

PONE-D-20-03333R2 

Child defecation and feces management practices in rural Bangladesh: Associations with fecal contamination, observed hand cleanliness and child diarrhea 

Dear Dr. Islam:

I'm pleased to inform you that your manuscript has been deemed suitable for publication in PLOS ONE. Congratulations! Your manuscript is now with our production department. 

Kind regards, 

on behalf of

Dr. William Joe 

Academic Editor

PLOS ONE